# CRT-QA: A Dataset of Complex Reasoning Question Answering over Tabular Data

**Zhehao Zhang**[1]*, **Xitao Li**[2]*, **Yan Gao**[3], **Jian-Guang Lou**[3]

[1]Dartmouth College, [2]Xi'an Jiaotong University

[3]Microsoft Research Asia

zhehao.zhang.gr@dartmouth.edu, wuitenye@stu.xjtu.edu.cn

{yan.gao, jlou}@microsoft.com

## Abstract

Large language models (LLMs) show powerful reasoning abilities on various text-based tasks. However, their reasoning capability on structured data such as tables has not been systematically explored. In this work, we first establish a comprehensive taxonomy of reasoning and operation types for tabular data analysis. Then, we construct a complex reasoning QA dataset over tabular data, named CRT-QA (**C**omplex **R**easoning QA over **T**abular data), with the following unique features: (1) it is the first Table QA dataset with multi-step *operation* and *informal reasoning*; (2) it contains fine-grained annotations on questions' directness, composition types of sub-questions, and human reasoning paths which can be used to conduct a thorough investigation on LLMs' reasoning ability; (3) it contains a collection of *unanswerable* and *indeterminate* questions that commonly arise in real-world situations. We further introduce an efficient and effective tool-augmented method, named ARC (**A**uto-exemplar-guided **R**easoning with **C**ode), to use external tools such as `Pandas` to solve table reasoning tasks without handcrafted demonstrations. The experiment results show that CRT-QA presents a strong challenge for baseline methods and ARC achieves the best result. The dataset and code are available at https://github.com/zzh-SJTU/CRT-QA.

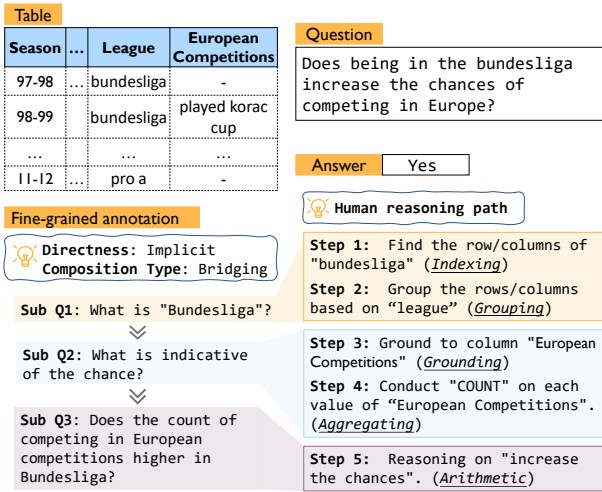

Figure 1: This example demonstrates the format of CRT-QA dataset: tables, question-answers, and fine-grained annotations.

Yao et al., 2023a) focus on LLMs' reasoning abilities on text-based NLP tasks. However, the capability of LLMs on table reasoning tasks has not been systematically investigated (Chen, 2023a). Evaluating LLMs' reasoning ability over tabular data and improving their performance can produce a significant impact on efficient data analysis, decision-making, and so on in real-life applications.

Current Table question answering (Table QA) datasets are primarily concerned with obtaining factoids to answer simple queries and lack in-depth analysis. Although recent works (Chen et al., 2021b) start to investigate multi-hop "reasoning" questions over tables, they do not have a clear definition of reasoning types and the "reasoning" they investigate (*e.g.*, operations like filtering) does not align with current research on LLMs' reasoning ability. Besides, the current Table QA datasets only contain *explicitly* questions. However, in real-life scenarios, users frequently ask *implicit* even *ambiguous* questions over tables.

To fill these gaps and conduct an in-depth anal-

## 1 Introduction

Large language models (LLMs) (Brown et al., 2020; Chowdhery et al., 2022; Chung et al., 2022; Touvron et al., 2023; OpenAI, 2023a,b) have recently shown emergent abilities, such as the capacity for "reasoning", when they are sufficient in size (Wei et al., 2022). A large number of works (Zhang et al., 2022a; Wei et al., 2023; Kojima et al., 2023;

---

*Work done during Zhehao and Xitao's internship at Microsoft Research Asia.

ysis of LLMs' reasoning abilities over tabular data, we first establish a fine-grained taxonomy of commonly-used *reasoning* and *operations* types for table analysis. Different from previous works (Chen et al., 2021b), we separate the steps that can be easily executed using a single `Pandas` or `SQL` query from reasoning and categorize them as *operations*. Following recent studies on the reasoning capacity of LLMs(Wei et al., 2023), we focus on *informal reasoning* which utilizes intuition, experience, and common sense to deduce outcomes.

Then, we construct CRT-QA dataset (**C**omplex **R**easoning QA over **T**abular data) over Wikipedia tables. Answer-based evaluation proves inadequate for assessing LLMs' reasoning ability, as it does not fully capture the complexity of their cognitive processes. Nonetheless, devising a robust method for evaluating such reasoning capabilities remains a formidable challenge within the field. When dealing with complex table analysis queries, humans typically begin by reformulating the questions (possibly implicitly) into more explicit ones, followed by decomposing them into sub-questions, and ultimately conducting atomic reasoning. Inspired by this process, we propose fine-grained annotations on the directness of questions, composition types of sub-questions, and human reasoning paths. To explore the ambiguous questions mentioned earlier, we incorporate a subset of *unanswerable and indeterminate* queries. During question collection, we propose a human-in-the-loop question generation pipeline that utilizes LLM to generate questions necessitating complex, multi-step reasoning. Our proposed pipeline can efficiently produce high-quality queries while mitigating issues such as biases, insufficient complexity, and lack of diversity.

We evaluate LLMs (e.g., GPT-4) with different prompting methods on CRT-QA. Inspired by the finding that LLMs can often generate correct reasoning plans but fail on execution, we propose an efficient and effective method, named ARC (**A**uto-exemplar-guided **R**easoning with **C**ode), to alleviate such limitation. Instead of expensive human effort for code design, ARC first uses an instructional prompt to generate exemplar code on the dev set queries and serve as an in-context demonstration for test questions. After executing the generated code with an external *Python* interpreter, we then inject the output into the prompt and LLM generates the final answer by reflection. Experiment results demonstrate that CRT-QA poses a signifi-

cant challenge for baseline methods, as the current most powerful model, GPT-4, achieves an accuracy of 56.32% through few-shot in-context learning. Our proposed ARC achieves the best result, outperforming various prompting and tool-use baselines.

## 2 Related Works

### 2.1 TableQA Datasets

Table QA is the task of answering queries concerning tabular data. A large number of datasets have been proposed for this task. Datasets such as WTQ (Pasupat and Liang, 2015), WikiSQL (Zhong et al., 2017), SQA (Iyyer et al., 2017) and Spider (Yu et al., 2018) contain tables for QA or text-to-SQL tasks. Recently, numerous works construct datasets that require multi-hop reasoning on tables: OT-TQA (Chen et al., 2021a), HybridQA (Chen et al., 2021b), TabFact (Chen et al., 2020b), LogicNLG (Chen et al., 2020a), AIT-QA (Katsis et al., 2021), MultiModalQA (Talmor et al., 2021), FeTaQA (Nan et al., 2021). However, they are focused on iterated factoid retrieval (Ho et al., 2022) where the definition of reasoning does not align with the reasoning ability of LLMs. Datasets like FinQA (Chen et al., 2022b), TAT-QA (Zhu et al., 2021), MultiHiertt (Zhao et al., 2022) and TABMWP (Lu et al., 2023b) focus on numerical reasoning over tabular data. Yin et al., 2022 propose ARCADE, a benchmark of 1,082 code generation using the pandas for tabular data analysis. However, they do not introduce commonsense in the datasets and their labels are not natural languages.

### 2.2 Language Models for Reasoning

**LLMs' reasoning abilities** Numerous works (Fu et al., 2023b; Wang et al., 2023b; Zelikman et al., 2022; Creswell et al., 2022; Yao et al., 2023a) focus on increasing LLM's arithmetic (Lewkowycz et al., 2022; Chen et al., 2022a; Zhou et al., 2022; Taylor et al., 2022), commonsense (Liu et al., 2022; Madaan et al., 2022) and symbolic reasoning (Zhou et al., 2023). Notably, simply adding "Let's think step by step" before each answer or using chain-of-thought (CoT) (Wei et al., 2023) prompting which contains a number of intermediate steps can better elicit LLM's reasoning ability.

**LLM with tools** External tools such as web browsers, search engines, Python interpreters, and models of other modalities have been incorporated to complete complex tasks (Nakano et al.,

| | Sub type | Example | Percentage |
|---|---|---|---|
| Operation | Indexing | *Which county* has had the fastest rate of population growth between 1960 and 2040, in terms of percentage change per decade? | 84.97% |
| | Filter | Did any drivers *who retired* due to an accident complete more laps than those who retired for other reasons? | 30.20% |
| | Grouping | Is there *a particular player* that Tom has faced more frequently than others? | 38.39% |
| | Sorting | Was there any significant increase or decrease in the number of points or wins for Juan Garriga *over the years*? | 5.64% |
| Reasoning | *Aggregating* | How many matches in the 1974-75 FA Cup tournament had a scoreless draw? | 58.12% |
| | *Arithmetic* | What is the average *time difference* between a manager's dismissal and the subsequent appointment of their replacement? | 29.26% |
| | Grounding | Does Juan Garriga have a higher *success* rate when competing with the Yamaha team compared to the JJ Cobas or Cagiva teams? Details: The term of *success* in the question is mapped to *win (a column)* | 17.99% |
| | Auto-categorization | What proportion of the Malaysia Airlines group companies are involved in the *airline industry*? Details :*airline industry* belongs to *principal activities (a column)* | 1.07% |
| | Temporal Reasoning | How did the Tampa Bay Buccaneers perform during *the first half of the 1983* season compared to *the second half of the season*? | 3.89% |
| | Geographical/-Spatial Reasoning | Does Andrea Petkovic have a higher winning percentage in finals matches held *in Europe* or *outside of Europe*? | 2.55% |
| | Reasoning with Quantifiers | Are there *any shows* that have been airing consistently throughout all networks in the Canadian Network Television Schedule in 1998-99? | 24.16% |
| | Others | Was there a *consistent difference* in the duration of operation between satellites launched earlier in the program and the later ones? | 50.20% |

Table 1: Our proposed taxonomy for operation and reasoning types in Table QA, accompanied by examples and their proportion in CRT-QA. We emphasize keywords for their respective categories.

2022; Shuster et al., 2022; Cheng et al., 2023; Cobbe et al., 2021; Paranjape et al., 2023; Shen et al., 2023; Lu et al., 2023a). Toolformer (Schick et al., 2023) uses self-supervision to teach LLMs to use multiple tools. However, it needs to fine-tune LLM's parameters, which makes it impractical to apply it to close-sourced LLMs like GPT-4. Yao et al., 2023b propose ReAct, a prompt-based paradigm to integrate reasoning and acting for LLMs. However, ReAct requires multiple API callings and hand-crafted exemplars of (*Thought*, *Act*, *Obs*) triplets, which has high calling cost and not flexible to transfer to other tasks.

## 3  CRT-QA Dataset

In this section, we describe the task formulation, and process for collecting tables, questions, answers, and detailed annotations for CRT-QA.

### 3.1  Desiderata

The table-based QA task is defined as the problem of generating an answer $a$ to a question $q$ based on a table $T$ with metadata $m$ using a model $M$ which can be formulated as $a = M(T, m, q)$. In our dataset,

The format of answers $a$ is free-form natural language. Our dataset focuses on the questions that require multiple steps of operation $\{o_1, o_2, ..., o_n\}$ and reasoning $\{r_1, r_2, ..., r_n\}$.

Previous TableQA datasets (Chen et al., 2021b) definition of "reasoning" primarily encompasses basic operations like filtering, which does not align with the more comprehensive understanding of reasoning in current LLM research, which involves higher-order cognitive tasks such as logical, numerical, and commonsense reasoning. As a result, we separate these steps from reasoning types and define them as operations. Following recent works on LLMs' reasoning ability (Wei et al., 2023; Cobbe et al., 2021), we examine *informal reasoning*, which relies on intuition, experience, and common sense to draw conclusions and solve problems. Inspired by benchmarks such as Big-bench (Srivastava et al., 2022), we propose a taxonomy on fine-grained reasoning types commonly used in table analysis. The operation and reasoning types are illustrated in Table 1.

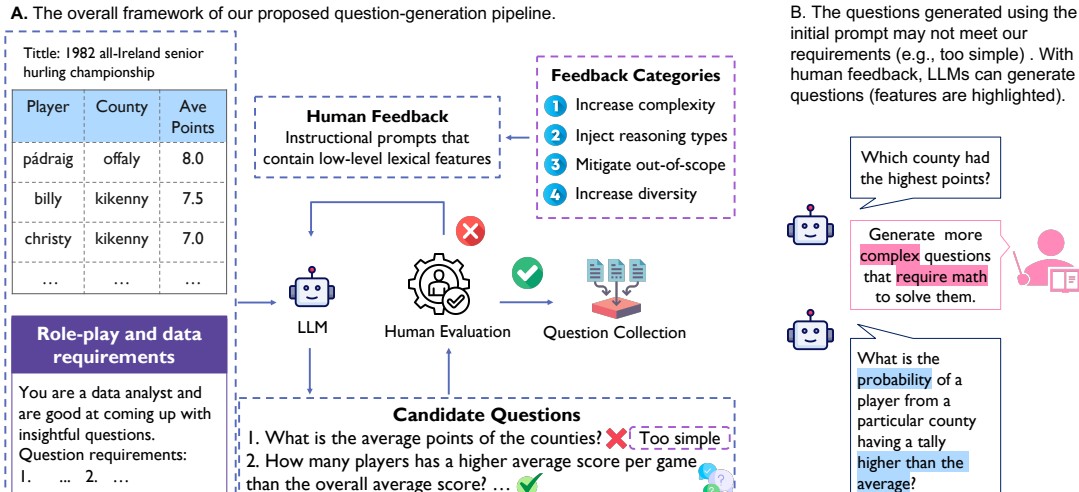

Figure 2: A: Our proposed pipeline of human-in-the-loop question generation using LLMs. we first design a role-playing prompt as a data analyst with desired questions' requirements for initial question generation. Then, human annotators collect the questions that meet the requirements and provide feedback for question improvement. Our proposal can efficiently collect high-quality and diverse table-based questions. B: A example of interaction between LLMs and human feedback. As we can see, specific feedback (e.g., *complex* and *math*) can greatly improve the quality of our generated questions.

## 3.2 Dataset Collection

We select open-domain tables from the TabFact (Chen et al., 2020b) datasets, where the tables are from Wikipedia[1]. Then, inspired by recent works on LLM's ability to aid human annotations (Bartolo et al., 2022; Törnberg, 2023), we design a pipeline to efficiently generate multi-step complex reasoning questions by incorporating LLMs and human feedback. After obtaining the questions, we conduct fine-grained annotations on their directness, decomposition types, and human reasoning paths.

### 3.2.1 Human-in-the-loop question generation using LLMs

As shown in Figure 2, the pipeline has two main steps: initially generating queries using LLMs, followed by human selection and feedback to enhance them in accordance with human preferences.

**Initial question generation**  Inspired by the effectiveness of LLMs' role-playing capability (Park et al., 2023; Wang et al., 2023a; Fu et al., 2023a; Liu et al., 2023), we use LLM (i.e., ChatGPT ) as the question generator, which largely reduces the cost of data annotations. Specifically, we design an instructional prompt containing question requirements to generate question candidates. However, there are three problems when we use such prompts for ChatGPT: **(i). lack of complexity**: Although we provide corresponding instructions on complex-

ity, ChatGPT usually generates simple questions that do not contain multi-hop reasoning; **(ii). lack of diversity**: When we ask ChatGPT to generate multiple questions, we find that many queries have similar formats. For example, the majority of them start with '*Is there*'; **(iii). unanswerable questions**: ChatGPT may generate questions that can not be answered only given the table. We collect some unanswerable and indeterminate questions and conduct an in-depth analysis in Section 6. The next paragraph described the approach we use to mitigate the above issues.

**Human selection and feedback**  Human feedback is essential for LLMs because it helps them align with human preferences and values. Inspired by recent works on model refinement (Ouyang et al., 2022; Huang et al., 2022; Shinn et al., 2023), we let human annotators select the questions that meet our requirements and then provide LLM with feedback to improve the quality of the questions. For feedback design, we use several lexical features such as *use math* and *more complex* to resolve the problems mentioned above and reduce potential biases. Empirically, we find that ChatGPT can better improve their generated questions by providing them with specific lexical features than high-level instructions. Details on the feedback design can be found in Appendix A.1.

---

[1]https://www.wikipedia.org/

### 3.2.2 Fine-grained annotations

Among the reasoning datasets, most of them only contain label-related annotations without human reasoning paths or fine-grained reasoning types. However, we argue that only goal-oriented annotations are insufficient to analyze the reasoning ability of LLMs. To fill in this gap, after annotating the answer, we further annotate whether a question is implicit or explicit and how sub-questions are composed. We also annotate the main steps of table operations and reasoning. After that, we use a template-filling method to efficiently annotate human reasoning paths to solve the questions. The details on template design and the complete annotation interface can be found in Appendix E and F.

**Directness**   Inspired by StrategyQA (Geva et al., 2021), we first introduce implicit questions over tabular data. Following Geva et al., 2021, we use the following rule-of-thumb to determine whether a question is implicit or explicit: the question is explicit if it can be written using words from the question, their inflections, and function words, while implicit questions require new content words to describe the reasoning process.

**Decomposition types**   As the queries in our dataset contain multi-step reasoning, we further annotate how these sub-questions are composed together. Following Min et al., 2019, we categorize the question decomposition into the following 3 types[2]: **bridging** needs to find the first-hop evidence in order to find the second-hop evidence; **intersection** requires finding an entity that meets two independent requirements; **comparison** requires comparing the property of two different entities. Our annotation can be used to analyze LLMs' question decomposition abilities.

**Human reasoning path**   To better evaluate LLMs' reasoning ability, we further annotate human reasoning paths for solving these queries. However, it is impractical for annotators to write their detailed reasoning paths due to the great volume of data. Hence, we design a template-filling paradigm to let annotators fill the objects of reasoning or operation. We first let annotators select the type of reasoning or operation for each step in order (selections are listed in Table 1). Then, for each step, they are asked to fill in a template

| Property | Value |
|---|---|
| Unique Tables | 423 |
| Total Questions | 1000 |
| Answerable Questions | 744 |
| Unanswerable Questions | 256 |
| Question Length (Avg/Median)♥ | 141.2 / 144.5 |
| Answer Length (Avg/Median) | 5.5 / 3.0 |
| Annotation Length (Avg/Median)♥ | 54.3 / 45.0 |
| Rows per Table (Avg/Median) | 12.6 / 10.0 |
| Num of reasoning (Avg/Median)♥ | 3.2 / 3.3 |
| Num of operation (Avg/Median)♥ | 3.1 / 2.8 |
| Length of reasoning path (Avg/Median) ♥ | 2.9 / 3.0 |
| Complexity (Agreement)† | 4.1 (88%) |
| Inter-annotator Agreement‡ | 93.7% |

Table 2: Core Statistics of CRT-QA. Lengths are the number of characters. Both ♥ and the complexity (assessed by humans) demonstrate the significant challenge posed by our dataset.

that specifies the objectives of the chosen type. For example, if *Aggregation* is involved in solving the query, the annotator should select which type of this aggregation (e.g., *sum*) and its objectives (e.g., *column names*). The template can be found in Table 8 in the Appendix. This method can efficiently annotate the key reasoning steps of humans.

### 3.3 Dataset Analysis and Statistics

**Key statistics**   Table 2 shows the key statistics of CRT-QA. Following Nan et al., 2021, we ask human annotators to rate the complexity of 100 samples on a scale of 1 to 5 and report the average rating and agreement. We compute the inter-annotator agreement of 3 annotators on 100 samples‡. The large proportion of agreement (92.7%) indicates the annotation quality of our dataset.

**Question and topic types**   We show the distribution of topics and question types within CRT-QA through visualizations. Due to page limitations, we have included them and their details in Appendix C and D. These visualizations show that our dataset encompasses a diverse array of topics and includes a variety of question types, ensuring comprehensive coverage and versatility in the QA domain.

**Comparison to existing datasets**   Compared with previous datasets[3], CRT-QA dataset exhibits several distinctive features: (1) CRT-QA is the first Table QA dataset that contains both multi-step operations and informal reasoning. (2) CRT-QA is the first Table QA dataset that contains fine-grained annotations on questions' directness, composition types, and human reasoning paths. (3) CRT-QA

---

[2]Examples of these 3 decomposition types are in Appendix B

[3]Table 11 in the Appendix comprehensively compare them.

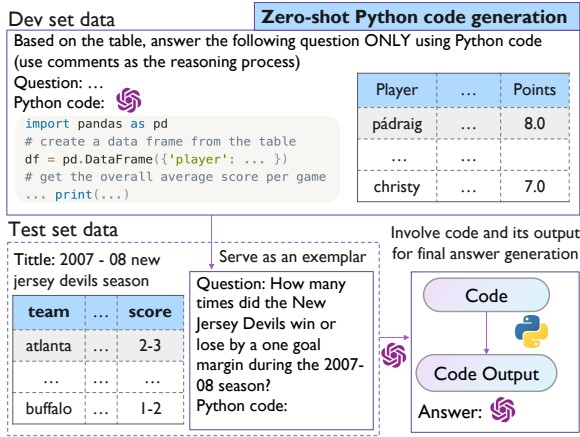

Figure 3: Our proposed ARC for complex table reasoning tasks. We first sample a data instance from the dev set and input it with an instructional prompt to LLMs for zero-shot code generation. We then use the generated code as the in-context exemplar to generate code for test data. After that, we execute the code and inject the output to the prompt, and iteratively input LLM to get the final answer. The prompts and dev-set demonstration can be found in Appendix. ARC can mitigate the shortcoming of LLMs for operation/reasoning execution and eliminate the effort of handcrafted code demonstrations.

has a sub-set of unanswerable and indeterminate questions, which are frequently occurred due to the complexity of real-life scenarios.

## 4 Method

Although LLMs show powerful reasoning abilities on various tasks (Qin et al., 2023), they have limitations on various fields (Lewkowycz et al., 2022; Ziems et al., 2023). From our pivot experiments of prompting baselines, we find that LLMs can often generate correct reasoning plans but are unable to appropriately execute them. However, such steps (e.g., arithmetic, counting) can be perfectly performed by external tools such as SQL or Pandas. Inspired by recent works on tool-augmented LLMs (Gao et al., 2023; Yao et al., 2023b; Shen et al., 2023; Lu et al., 2023a; Paranjape et al., 2023), we propose an efficient and effective approach, named ARC (**A**uto-exemplar-guided **R**easoning with **C**ode), to use external tools such as Pandas to solve table reasoning tasks without handcrafted demonstrations. Figure 3 illustrates the pipeline of our proposed ARC and the detailed prompt design for each step can be found in Appendix H.

**Auto-exemplar generation** Although manually-annotated demonstration shows significant effectiveness in in-context learning for LLMs, nontrivial hand-drafting of effective exemplars makes it not flexible enough to be applied to sophisticated tasks

such as code generation for complex table analysis. Inspired by recent works on auto-demonstration generation (Zhang et al., 2022b), we first randomly sample a data instance from the development set and input LLMs with an instructional prompt for code generation. The prompt we use is a simple instruction to generate Python code and print intermediate or final results. The ablation of different selections will be discussed in Appendix H.

**In-context code generation** For every data example in the test set, we use the exemplar generated from the dev set to conduct in-context learning for code generation. As Pandas is the most commonly-used library in Python for tabular data analysis which may frequently occur in LLMs' pretraining data, the generated codes are proficient in the use of Pandas to process tables.

**Code execution with external tools** We further use the generated code for execution using a Python interpreter with a Pandas installed environment. We then obtain the output of the program as the intermediate or final results for the query.

**Iterative LLM calling with code output** However, for queries that require in-depth common-sense reasoning, only the code sometimes can not directly solve them. As a result, inspired by Re-Act (Yao et al., 2023b), we also integrate *Acting* and *Reasoning* by injecting code output into the prompt design for final step reasoning. By prompting LLMs with code output, LLMs can generate more accurate final answers by avoiding step execution errors.

## 5 Experiments

### 5.1 Experiment Settings

For all the experiments, we use the powerful Chat-GPT, GPT-3.5-turbo, and GPT-4 as the LLMs to investigate their reasoning ability on tabular data. Following Chen, 2023b, we use Markdown as the format of tables and use Exact Match (EM) as the main metric for evaluations. For each experiment, we run three times with different random seeds and report the average EM score.

### 5.2 Baselines

To evaluate LLMs' reasoning ability on the complex TableQA task, we select the following baselines:[4] **Few-shot/Zero-shot prompting** (Brown

---

[4]Implementation details can be found in Appendix G

| Method | Operation Types | | | | Reasoning Types | | | | | | | | Overall |
|---|---|---|---|---|---|---|---|---|---|---|---|---|---|
| | Index | Sort | Group | Filter | GRO | CAT | TEM | AGG | ARI | SPA | QUA | OTH | |
| *Prompting w/ ChatGPT* | | | | | | | | | | | | | |
| Zero-shot | 47.20 | 50.00 | 46.34 | 37.15 | 46.34 | 44.44 | 53.09 | 36.39 | 37.68 | 42.11 | 62.22 | 52.41 | 46.11 |
| Zero-shot-CoT | 40.22 | 40.00 | 48.07 | 32.38 | 28.37 | 44.44 | 44.44 | 27.92 | 33.61 | 21.05 | 50.56 | 37.70 | 37.23 |
| Few-shot (2-shot) | 47.48 | 37.57 | 55.44 | 50.00 | 42.31 | 40.74 | 56.79 | 34.73 | 37.54 | 42.11 | 51.80 | 65.56 | 45.92 |
| Few-shot-CoT (2-shot) | 46.45 | 46.67 | 53.10 | 38.80 | 44.21 | 37.04 | 37.04 | 36.39 | 40.48 | 24.56 | 62.41 | 50.96 | 45.47 |
| *Tool Use w/ ChatGPT* | | | | | | | | | | | | | |
| PAL | 46.51 | 42.96 | 51.27 | 36.39 | 42.41 | 11.11 | 42.59 | 35.54 | 41.87 | 17.54 | 63.38 | 45.37 | 44.11 |
| ReAct | 47.69 | 58.97 | 51.58 | 38.21 | 43.57 | 22.22 | 44.44 | 42.08 | 22.22 | 46.77 | 60.00 | 46.77 | 45.24 |
| **ARC (Ours)** | 50.62 | 43.33 | 58.20 | 40.42 | 47.15 | 37.04 | 37.74 | 44.19 | 45.46 | 40.35 | 64.83 | 51.15 | 49.41 |
| *Prompting w/ GPT-3.5-turbo* | | | | | | | | | | | | | |
| Zero-shot | 43.83 | 45.00 | 50.18 | 34.02 | 39.01 | 44.44 | 55.56 | 31.93 | 31.51 | 10.53 | 63.78 | 48.97 | 42.11 |
| Zero-shot-CoT | 43.84 | 55.00 | 52.58 | 33.20 | 39.01 | 33.33 | 37.04 | 34.80 | 33.19 | 26.32 | 52.78 | 44.37 | 41.58 |
| Few-shot (2-shot) | 49.69 | 47.50 | 62.40 | 46.10 | 55.56 | 59.26 | 38.62 | 43.70 | 43.11 | 42.11 | 66.67 | 54.48 | 49.05 |
| Few-shot-CoT (2-shot) | 47.20 | 45.00 | 56.49 | 37.30 | 45.39 | 44.44 | 33.33 | 37.86 | 39.92 | 53.63 | 62.22 | 51.26 | 46.33 |
| *Tool Use w/ GPT-3.5-turbo* | | | | | | | | | | | | | |
| PAL | 53.19 | 50.00 | 61.59 | 43.03 | 50.35 | 11.11 | 51.85 | 46.07 | 54.81 | 31.58 | 67.22 | 51.38 | 52.17 |
| ReAct | 42.71 | 35.00 | 45.96 | 34.42 | 38.30 | 22.22 | 40.74 | 33.07 | 36.55 | 21.05 | 52.77 | 42.76 | 40.22 |
| **ARC (Ours)** | 55.28 | 52.50 | 64.21 | 51.77 | 66.67 | 60.71 | 46.56 | 46.56 | 52.12 | 16.67 | 63.53 | 54.38 | 53.26 |
| *Prompting w/ GPT-4* | | | | | | | | | | | | | |
| Zero-shot | 46.99 | 51.28 | 56.47 | 35.27 | 50.46 | 11.11 | 48.15 | 34.92 | 39.74 | 36.84 | 60.23 | 55.16 | 46.21 |
| Zero-shot-CoT | 42.00 | 43.59 | 50.00 | 37.29 | 39.45 | 22.22 | 37.03 | 32.14 | 36.32 | 31.58 | 53.80 | 45.08 | 41.01 |
| Few-shot (2-shot) | 57.75 | 43.59 | 64.39 | 55.05 | 66.67 | 55.56 | 44.25 | 44.25 | 51.71 | 36.84 | 78.95 | 63.31 | 56.32 |
| Few-shot-CoT (2-shot) | 59.29 | 41.03 | 67.63 | 50.00 | 56.88 | 66.67 | 51.85 | 49.01 | 54.70 | 52.63 | 77.19 | 61.15 | 58.69 |
| *Tool Use w/ GPT-4* | | | | | | | | | | | | | |
| PAL | 61.20 | 48.72 | 65.47 | 52.97 | 56.88 | 33.33 | 40.74 | 54.56 | 62.39 | 36.84 | 74.85 | 60.67 | 59.83 |
| ReAct | 61.88 | 65.00 | 56.94 | 45.00 | 56.94 | 75.00 | 44.44 | 46.95 | 54.69 | 57.14 | 76.86 | 61.34 | 58.69 |
| **ARC (Ours)** | 62.14 | 64.10 | 64.75 | 51.06 | 54.13 | 55.56 | 55.56 | 52.09 | 59.40 | 52.63 | 72.94 | 65.16 | 60.11 |

Table 3: Evaluation results of various baselines and our method on our proposed CRT-QA: **GRO**: Grounding; **CAT**: Auto-categorization; **TEM**: Temporal reasoning; **AGG**: aggregating; **ARI**: Arithmetic; **SPA**: Spatial/ Geographical reasoning; **QUA**: Reasoning with quantifiers; **OTH**: Other commonsense reasoning. The mean *p*-values for the paired *t*-test between ARC and other top-performing baselines is 0.041, indicating significant differences. Among all the methods except Zero-shot and Zero-shot-CoT, ARC is the **Only** method that requires no handcrafted exemplar.

et al., 2020): simply prompts LLMs with few-shot examples or instructions. **Few-shot/Zero-shot CoT** (Wei et al., 2023; Kojima et al., 2023): inputs LLMs few-shot exemplars with manually-crafted reasoning path or *Let's think step by step*. **PAL** (Gao et al., 2023): uses few-shot examples of only Python code to encourage LLMs to generate correct code for problem-solving. **ReAct** (Yao et al., 2023b) utilizes in-context examples of (*Thought*, *Act*, *Obs*) tuples to combine logical path and task-specific actions. **ARC** utilizes a zero-shot-generated code exemplar to perform in-context code generation and incorporate code output for final answer generation.

## 5.3 Experiment Results

Table 3 shows different methods' EM scores on CRT-QA dataset. We can see that (1). overall, the most efficacious approach achieves a maximum of 60.11, indicating the difficulty of our dataset. (2). among all the baselines, our proposed ARC achieves the **best** average EM scores with an average improvement of 1.846 across all models without using any handcrafted exemplar, indicating the effectiveness of our proposal. For ChatGPT baselines. (3). we find that Zero-shot-CoT performs even worse than the vanilla Zero-shot approach. By checking the reasoning paths elicited by *Let's think step-by-step*, we find that the reason may arise from the phenomenon that the reasoning paths are unruly and even generate codes that the model does not have the ability to solve. As a result, *Let's think step-by-step* is not a one-fits-for-all solution. (4). although Few-shot-CoT can not outperform Zero-shot for ChatGPT. As the model evolves (i.e., from ChatGPT to Turbo to GPT-4), Few-shot-CoT can have better performances than Zero-shot predictions, indicating that the model increases its

CoT reasoning ability. (5). Among the 3 tool-use baselines, ReAct can not have comparative performances with the other two methods with GPT-3.5-turbo and GPT-4. By investigating the reasoning path, we find that ReAct often finishes without any answer. Alternatively, ReAct often conducts a substantial number of iterations, resulting in not only increased costs but also an extremely long reasoning pathway that becomes out of control.

From fine-grained reasoning types shown in Table 3, we observe that all prompting-based methods are bad at *aggregation* and *arithmetic* compared with other reasoning types. Noticeably, our proposed ARC and PAL can greatly improve LLMs' ability on these two reasoning types. Besides, we observe that among all the reasoning types, LLMs perform the best in reasoning with quantifiers. Due to page constraints, a comprehensive ablation study on the number of exemplars, error analysis, and case study are in Appendix H, J, and I.

## 6 Unanswerable and Indeterminate Question

Most Table QA datasets are designed for answering the questions with golden labeling (Pasupat and Liang, 2015; Chen et al., 2020b, 2021b,a), but real users possibly ask questions that are inherently difficult to answer due to the complexity of the real world. Motivated by this, we incorporate a sub-set of *unanswerable* and *indeterminate* queries where some questions go beyond common external knowledge, while others are inherently problematic. We categorize these questions into four categories and conduct the answerability of LLMs based on them.

| Type | Definition | Percentage |
|------|-----------|-----------|
| Out of scope | Lacking essential information based on the given table. | 73.2% |
| Hallucination | The assumption in the question is invalid based on the table. | 10.4% |
| Problematic | Question itself contains logical errors. | 4.8% |
| Subjective | The answer varies from annotators due to different metrics, algorithms, and criteria. | 6.4% |
| Others | Other types of questions that can not be labeled | 5.2% |

Table 4: Category and ratio of indeterminate and unanswerable questions in CRT-QA dataset.

As shown in Table 4, out-of-scope, hallucination, and problematic questions are **unanswerable**. The main reason is the absence of essential information or logical flaws within the question itself. For example, there is an implicit assumption underlying

the question "If the score increase by years, ...", but the table content cannot support the implied assumption. For **indeterminate questions**, annotators can yield different answers due to different metrics, algorithms, and criteria. These questions can be answered from both subjective and objective perspectives. A "best guess" can be made using subjective reasoning, while these questions can also be objectively asked for user's further clarification in certain scenarios.

**Answerability Study** We evaluate the language model's ability to determine whether to answer a question under three approaches. As a baseline approach, **Random** approach randomly predicts the responses under the prior probability distribution. **Binary Classification** presents a binary classification problem, wherein the model must output with either "unanswerable" or "answerable". **Question Answering** approach produces the correct answer if the question is answerable and respond with "unanswerable" otherwise. We conduct these experiments on the whole dataset, including unanswerable/indeterminate and "normal" questions.

| | Acc | P | R | F1 |
|---|-----|---|---|-----|
| Random | 0.596 | 0.731 | 0.727 | 0.729 |
| Binary Classification | 0.680 | 0.908 | 0.637 | 0.749 |
| Question Answering | **0.779** | **0.928** | **0.763** | **0.838** |

Table 5: Results for identifying answerability. We report common metrics for binary classification, i.e., Acc (Accuracy), P( Precision), R(Recall) and F1 score.

Based on the results presented in Table 5, Binary Classification shows improvements over Random, indicating its effectiveness in identifying questions' answerability. Question Answering proves to be the most effective approach for identifying answerability, probably because generating answers is easier than determining whether a question can be answered. It mimics some pre-training tasks like reading comprehension. This study benefits a broader understanding of how language models can tackle unanswerable and indeterminate questions and provides directions to enhance performance.

## 7 Conclusion

In this work, to systematically evaluate LLMs' reasoning ability on tabular data, we first establish a comprehensive taxonomy on operation and reasoning types for table analysis. Then, we propose CRT-QA, a dataset of complex reasoning QA over

tables. We propose ARC which effectively utilizes table analysis tools to solve table reasoning tasks without manually-annotated exemplars. Extensive experiments show CRT-QA poses a significant challenge for LLMs and our proposed ARC achieves the best EM scores. Besides the main experiments, we also conduct thorough ablation studies, error analyses, answerability study, and case study for further analysis.

## Limitations

(1) CRT-QA is a test-only dataset, which means no gradient updates are performed. While striving for problem complexity, we face challenges in balancing the quantity of our dataset. This is primarily due to the intricate nature of our annotation process, which demands more time for answer generation and fine-grained process labeling. (2) Similar to previous works discussed in Table 11, we only foucs on single-table question answering. However, queries across multi-tables are also common in real-life table analysis scenarios. (3) We don't research the boundary of external knowledge. The appearance of unanswerable and indeterminate questions is associated with our data generation goal, which is to generate complex and diverse questions. Specifically, the indeterminate questions are contrasted with implicit questions, while indeterminate questions stand out beyond implicit questions. We leave this study as future work. (4) In our study, we utilize a combination of exact match and human evaluation as our evaluation metric. It is reasonable because, during the question generation process, we only select the questions that can be answered within several words without ambiguity. Although this is not comprehensive for free-form answer-generation tasks, alternative metrics such as F1, ROUGE-L, and BLEU-1 also possess inherent limitations. Evaluation of the free-form answer-generation task seems promising. Moreover, our fine-grained annotations provide a feasible path to help answer the question. Although the path is not unique. Currently, there is no effective method to evaluate the reasoning path. This aspect will be left for future research and development.

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

# A Prompt Design

| Hyper-parameter | Value |
|---|---|
| Temperature | 0.7 |
| max_len (CoT) | 1024 |
| max_len (Code) | 1024 |
| max_len (Few-shot/Zero-shot) | 16 |
| top_p | 1.0 |
| best_of | 1 |

Table 6: Hyper-parameter setting for LLMs.

| Selection from dev set | Average EM score |
|---|---|
| Selection 1 | 49.41 |
| Selection 2 | 49.89 |
| Selection 3 | 48.96 |

Table 7: Different dev set selections' performance for ARC.

### A.1 Human feedback for question generation

The followings are examples of feedback for LLMs to generate desired questions.

- Generate another 10 more complex questions.

- Generate another 10 questions with different question types.

- Generate another 10 more complex questions that require math to solve them.

- Generate another 10 more complex questions that require common sense for column 1. Other choices may also improve the quality of LLMs' generated questions.

## A.2 Prompt for Baselines

All prompt designs for the main experiment and experiment in Section 6 can be found in Table 8 and Table 9 respectively.

## B Question Decomposition Types

Following Min et al., 2019, we study the following three different types of question decomposition types:

- **Bridging**: requires finding first-hop evidence before moving on to the second-hop evidence. Example question: "What was the average number of years between a TV station's affiliation with the e! Canadian TV system and their eventual disaffiliation?".

- **Intersection** requires finding an entity that meets two independent conditions. Example question: "Are there any counties within the Mid-Indiana Football Conference that contain more than one school?".

- **Comparison** requires comparing the features of two distinct entities. Example question: "How often does Tim Lajcik win fights in the first round compared to subsequent rounds?".

## C Data Topic Distribution

Following Parikh et al., 2020, we use Wikimedia Foundation's topic categorization model (Asthana and Halfaker, 2018) to visualize the topic distribution of our dataset. Figure 4 shows that our data are mostly related to sports, biography, regions, and media. Overall, CRT-QA dataset covers a fairly wide range of topic domains.

## D Question Type Distribution

Following (Yang et al., 2018), by taking the three neighboring tokens along with the central question word (CQW), we can determine the question types. A visual representation of the distribution is shown in Figure 5, which illustrates the syntactic diversity of questions in our proposed CRT-QA.

## E Data Annotation Interface

Figure 8 shows the detailed interface for data annotation.

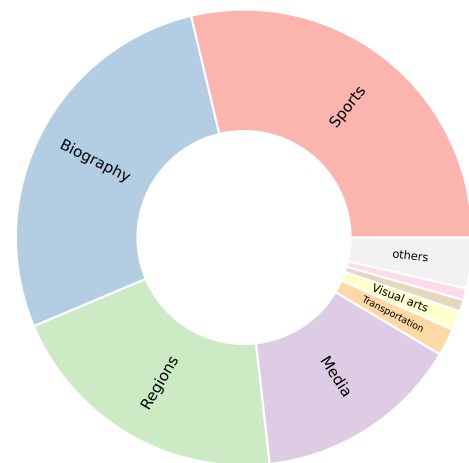

Figure 4: Topic distribution in CRT-QA. Categories in the figure originate from the mid-level WikiProjects directory.

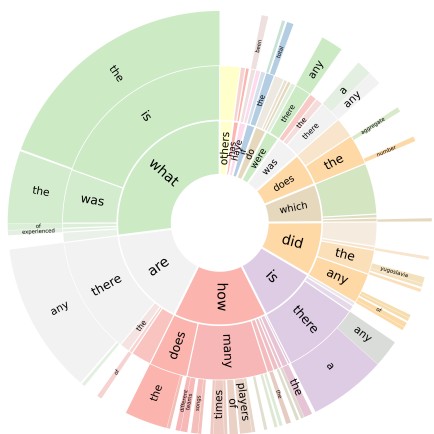

Figure 5: Types of questions in CRT-QA. Only high-frequency words are labeled, and empty blocks indicate that the frequencies of the suffixes are too rare to be shown individually

## F Data Annotation Details

We enroll 2 undergraduate students and 1 Ph.D. student majoring in computer science for data annotations. All of them have at least one year of data analysis experience.

## G Experiment Implementation Details

The models we use for experiments are `text-chat-davinci-003`, `GPT-3.5-turbo`, and `GPT-4` through Microsoft Azure API. For tool-use baselines, empirically, we find that the LLM-generated code may contain some syntax errors which make it impossible to run and generate output. For these cases, we let LLM re-generate code a maximum of five times. Once it

**zero-shot**

| | | |
|---|---|---|
| Table | Read the table below regarding "yugoslavia national football team results" | |

|   | date         | city            | opponent   | results | type of game |
|--:|:-------------|:----------------|:-----------|:--------|:-------------|
| 0 | april 18     | belgrade        | france     | 1:0     | 1966 wcq     |
| 1 | may 9        | belgrade        | england    | 1:1     | friendly     |
| 2 | june 16      | oslo , norway   | norway     | 0:3     | 1966 wcq     |
| 3 | september 4  | moscow , russia | ussr       | 0:0     | friendly     |
| 4 | september 19 | luxembourg      | luxembourg | 5:2     | 1966 wcq     |
| 5 | october 9    | paris , france  | france     | 0:1     | 1966 wcq     |
| 6 | november 7   | belgrade        | norway     | 1:1     | 1966 wcq     |

Question  Did the Yugoslavia national football team play any games against teams outside of Europe in the table? Answer with only 'Yes' or 'No' that is most accurate and nothing else.

Answer

**zero-shot-CoT**

Table  Read the table below regarding "yugoslavia national football team results"

|   | date         | city            | opponent   | results | type of game |
|--:|:-------------|:----------------|:-----------|:--------|:-------------|
| 0 | april 18     | belgrade        | france     | 1:0     | 1966 wcq     |
| 1 | may 9        | belgrade        | england    | 1:1     | friendly     |
| 2 | june 16      | oslo , norway   | norway     | 0:3     | 1966 wcq     |
| 3 | september 4  | moscow , russia | ussr       | 0:0     | friendly     |
| 4 | september 19 | luxembourg      | luxembourg | 5:2     | 1966 wcq     |
| 5 | october 9    | paris , france  | france     | 0:1     | 1966 wcq     |
| 6 | november 7   | belgrade        | norway     | 1:1     | 1966 wcq     |

Question  Did the Yugoslavia national football team play any games against teams outside of Europe in the table? Answer with only 'Yes' or 'No' that is most accurate and nothing else.

Answer  Let's think step-by-step

**1-shot**

Table  Read the table below regarding "1982 all - ireland senior hurling championship" to answer the following questions.

|   | rank | player            | county   | tally  | total | matches | average |
|--:|:----:|:------------------|:---------|:-------|:------|--------:|--------:|
| 0 | 1    | pádraig horan     | offaly   | 5 - 17 | 32    | 4       | 8       |
| 1 | 2    | billy fitzpatrick | kilkenny | 2 - 24 | 30    | 4       | 7.5     |
| 2 | 3    | tony o 'sullivan  | cork     | 0 - 28 | 28    | 4       | 7       |
| 3 | 4    | p j molloy        | galway   | 3 - 11 | 20    | 2       | 10      |
| 4 | 5    | christy heffernan | kilkenny | 3 - 9  | 18    | 4       | 4.5     |
| 5 | 5    | pat horgan        | cork     | 0 - 18 | 18    | 4       | 4.5     |

Question  How many players in the 1982 all-Ireland senior hurling championship had a higher average score per game than the overall average score per game of the competition?

Answer  4

Table  Read the table below regarding "yugoslavia national football team results"

|   | date         | city            | opponent   | results | type of game |
|--:|:-------------|:----------------|:-----------|:--------|:-------------|
| 0 | april 18     | belgrade        | france     | 1:0     | 1966 wcq     |
| 1 | may 9        | belgrade        | england    | 1:1     | friendly     |
| 2 | june 16      | oslo , norway   | norway     | 0:3     | 1966 wcq     |
| 3 | september 4  | moscow , russia | ussr       | 0:0     | friendly     |
| 4 | september 19 | luxembourg      | luxembourg | 5:2     | 1966 wcq     |
| 5 | october 9    | paris , france  | france     | 0:1     | 1966 wcq     |
| 6 | november 7   | belgrade        | norway     | 1:1     | 1966 wcq     |

Question  Did the Yugoslavia national football team play any games against teams outside of Europe in the table? Answer with only 'Yes' or 'No' that is most accurate and nothing else.

Answer

**2-shot**

Table  Read the table below regarding "1982 all - ireland senior hurling championship" to answer the following questions.

|   | rank | player            | county   | tally  | total | matches | average |
|--:|:----:|:------------------|:---------|:-------|:------|--------:|--------:|
| 0 | 1    | pádraig horan     | offaly   | 5 - 17 | 32    | 4       | 8       |
| 1 | 2    | billy fitzpatrick | kilkenny | 2 - 24 | 30    | 4       | 7.5     |
| 2 | 3    | tony o 'sullivan  | cork     | 0 - 28 | 28    | 4       | 7       |
| 3 | 4    | p j molloy        | galway   | 3 - 11 | 20    | 2       | 10      |
| 4 | 5    | christy heffernan | kilkenny | 3 - 9  | 18    | 4       | 4.5     |
| 5 | 5    | pat horgan        | cork     | 0 - 18 | 18    | 4       | 4.5     |

Question  How many players in the 1982 all-Ireland senior hurling championship had a higher average score per game than the overall average score per game of the competition?

Answer  4

Table  Read the table below regarding "g.d. estoril praia" to answer the following questions.

|   | season    | competition       | round    | opponent         | home  | away  |
|--:|:----------|:------------------|:---------|:-----------------|:------|:------|
| 0 | 2013 - 14 | uefa europa league | 3q       | hapoel ramat gan | 0 - 0 | 1 - 0 |
| 1 | 2013 - 14 | uefa europa league | play - off | pasching        | 2 - 0 | 2 - 1 |
| 2 | 2013 - 14 | uefa europa league | group h  | sevilla          | 1 - 2 | -     |
| 3 | 2013 - 14 | uefa europa league | group h  | slovan liberec   | -     | 1 - 2 |
| 4 | 2013 - 14 | uefa europa league | group h  | freiburg         | -     | 1 - 1 |

Question:  Was there a correlation between GD Estoril Praia's performance in home games and away games during the 2013-14 UEFA Europa League competition?

Answer:  No

generates runnable code, we execute it and get the output. If the LLM can not generate runnable code five times, we keep the code in the prompt and set the output to "None". The hyperparameters we use can be found in Table 6.

## H  Ablation Study

For our proposed ARC, we select 3 different examples from the dev set to conduct zero-shot code generation as exemplars for the test set. Table 7 shows that the performance difference among 3 different selections is within 1 EM score demonstrating the robustness of our proposal.

We also design four sets of contrast experiments for the ablation study as Figure 6 shows. We find the table reasoning ability differs from the models. GPT4 is best and turbo performs on par with Chat-GPT. For the in-context learning, GPT4 benefits a lot from the increase in the number of demonstrations, but the increase is not significant for other models. We study the impact of up to 2-shot because structured tables consuming lots of tokens can easily break the input limitation.

As expected, different decomposition types vary in difficulty with bridging being the most challenging and comparison being the easiest. Besides, LLMs obtain similar performances on implicit and explicit questions in our Table QA dataset.

## I  Case Study

We show how our method ACR uses external tools to solve table reasoning tasks in Figure 7. The comments in ACR show the reasoning sketch and guide the generation of code. Using external tools enhances numerical computation compared with plain text reasoning. In contrast, CoT fails even with the right reasoning path.

## J  Error Analyses

To analyze how the error was caused, we randomly choose 50 samples and go depth into error analysis based on the performance of ARC.

We find five types of errors: (1) Code generation error (20%). The code is not executable and the output is none or an illegal type. (2) Gross error of reasoning (32%). The reasoning path deviates from the requirements of the user query. (3) Condition missing error (18%). The code framework has no problem in general, but some subtle conditions or operations are missed. (4) Format error (26%). The model return with an answer but can't be judged by the metric. To ease the trouble of format error, we augment the EM with human evaluation. (5) Refuse to answer (4%). The answerable query is regarded as an unanswerable question. The model's completion contain some expressions that refused to answer like "I am unable to write Python code for this question as the data does not provide information ...". We find that the ARC method of the program enhances numerical processing while weakening the semantic recognition of the text. So the method makes mistakes for the match of strings such as "n/a" and "n / a".

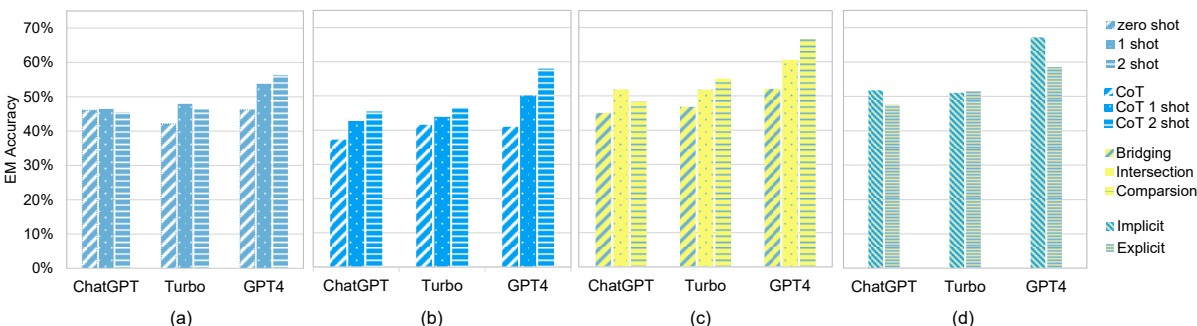

Figure 6: Ablation experiments in four groups: (a) number of demonstrations, (b) number of demonstrations for CoT, (c) decomposition types, (4) directness.

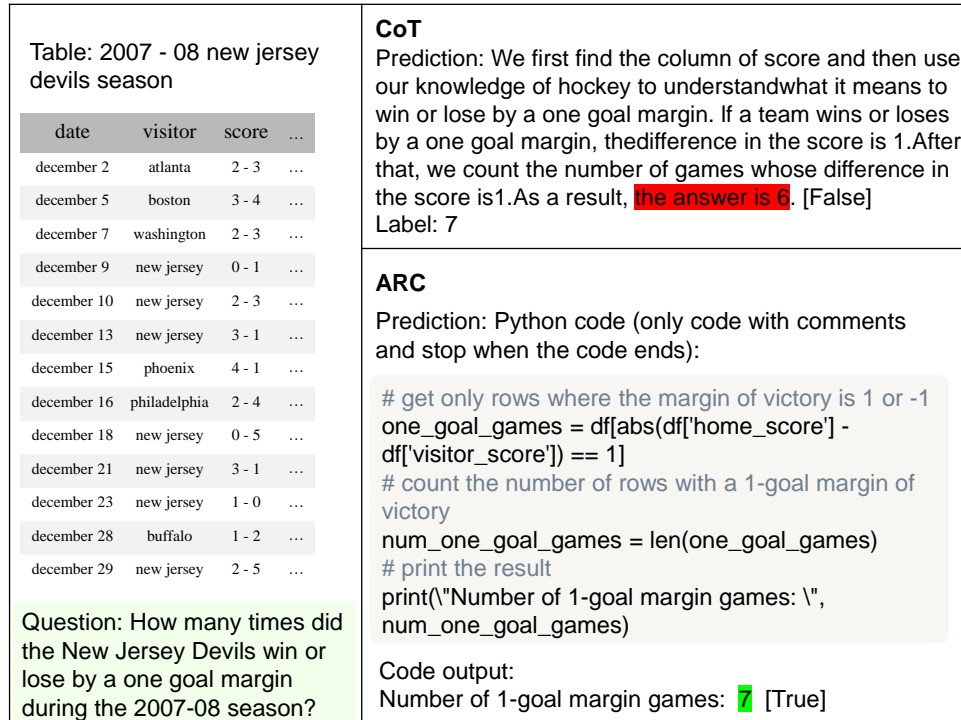

Figure 7: Case study on CoT and ARC. We can find the CoT can generate correct reasoning plans but fail on reasoning execution. On the contrary, ARC can obtain the right answer by using code.

| Dataset | Multi-hop Reasoning | Multi-hop Operations | Numerical Reasoning | Common Sense | Query Directness Implicit | Query Directness Explicit | Annotations Decomposition | Annotations Reasoning Path | Domain | Unanswerable Queries |
|---|---|---|---|---|---|---|---|---|---|---|
| WTQ (2015) | ✗ | ✓ | ✓ | ✗ | ✗ | ✓ | ✗ | ✗ | open | ✗ |
| TabFact(2020b) | ✓ | ✓ | ✓ | ✗ | ✗ | ✓ | ✗ | ✗ | open | ✗ |
| HybridQA (2021b) | ✓ | ✓ | ✓ | ✗ | ✗ | ✓ | ✗ | ✗ | open | ✗ |
| OTTQA (2021a) | ✓ | ✓ | ✓ | ✗ | ✗ | ✓ | ✗ | ✗ | open | ✗ |
| FinQA (2022b) | ✓ | ✓ | ✓ | ✗ | ✗ | ✓ | ✗ | ✗ | finance | ✗ |
| TAT-QA (2021) | ✓ | ✓ | ✓ | ✗ | ✗ | ✓ | ✗ | ✗ | finance | ✗ |
| AIT-QA (2021) | ✓ | ✓ | ✓ | ✗ | ✗ | ✓ | ✗ | ✗ | airline | ✗ |
| MultiHiertt (2022) † | ✓ | ✓ | ✓ | ✗ | ✗ | ✓ | ✗ | ✓ | finance | ✗ |
| FetaQA (2021) | ✓ | ✓ | ✓ | ✗ | ✗ | ✓ | ✗ | ✗ | open | ✗ |
| TABMWP (2023b) | ✓ | ✓ | ✓ | ✗ | ✗ | ✓ | ✗ | ✗ | open | ✗ |
| **CRT-QA (ours)** | ✓ | ✓ | ✓ | ✓ | ✓ | ✓ | ✓ | ✓ | open | ✓ |

Table 11: A comparison of CRT-QA and other Table QA datasets. CRT-QA is the first TableQA dataset that contains implicit questions, detailed annotations of human reasoning paths, and question decomposition types. †: MultiHiertt only contains math expression as the reasoning path.

| Table | Read the table below regarding "yugoslavia national football team results" |
|---|---|

|   | date | city | opponent | results | type of game |
|--:|:---|:---|:---|:--:|:---|
| 0 | april 18 | belgrade | france | 1:0 | 1966 wcq |
| 1 | may 9 | belgrade | england | 1:1 | friendly |
| 2 | june 16 | oslo , norway | norway | 0:3 | 1966 wcq |
| 3 | september 4 | moscow , russia | ussr | 0:0 | friendly |
| 4 | september 19 | luxembourg | luxembourg | 5:2 | 1966 wcq |
| 5 | october 9 | paris , france | france | 0:1 | 1966 wcq |
| 6 | november 7 | belgrade | norway | 1:1 | 1966 wcq |

| Question | Did the Yugoslavia national football team play any games against teams outside of Europe in the table? Answer with only 'Yes' or 'No' that is most accurate and nothing else. |
|---|---|
| Answer | |

---

**CoT 2-shot**

| Table | Read the table below regarding "1982 all - ireland senior hurling championship" to answer the following questions. |
|---|---|

|   | rank | player | county | tally | total | matches | average |
|--:|:--:|:---|:---|:---|:--:|:--:|:--:|
| 0 | 1 | pádraig horan | offaly | 5 - 17 | 32 | 4 | 8 |
| 1 | 2 | billy fitzpatrick | kilkenny | 2 - 24 | 30 | 4 | 7.5 |
| 2 | 3 | tony o 'sullivan | cork | 0 - 28 | 28 | 4 | 7 |
| 3 | 4 | p j molloy | galway | 3 - 11 | 20 | 2 | 10 |
| 4 | 5 | christy heffernan | kilkenny | 3 - 9 | 18 | 4 | 4.5 |
| 5 | 5 | pat horgan | cork | 0 - 18 | 18 | 4 | 4.5 |

| Question | How many players in the 1982 all-Ireland senior hurling championship had a higher average score per game than the overall average score per game of the competition? |
|---|---|
| Explanation | We first find the column of "average" and compute the average of all the players, which is (8 + 7.5 + 7 + 10 + 4.5 + 4.5)/6 = 6.917. Then we count the number of player whose average is larger than 6.917. As a result, the answer is 4. |
| Table | Read the table below regarding "g.d. estoril praia" to answer the following questions. |

|   | season | competition | round | opponent | home | away |
|--:|:---|:---|:---|:---|:---|:---|
| 0 | 2013 - 14 | uefa europa league | 3q | hapoel ramat gan | 0 - 0 | 1 - 0 |
| 1 | 2013 - 14 | uefa europa league | play - off | pasching | 2 - 0 | 2 - 1 |
| 2 | 2013 - 14 | uefa europa league | group h | sevilla | 1 - 2 | - |
| 3 | 2013 - 14 | uefa europa league | group h | slovan liberec | - | 1 - 2 |
| 4 | 2013 - 14 | uefa europa league | group h | freiburg | - | 1 - 1 |

| Question: | Was there a correlation between GD Estoril Praia's performance in home games and away games during the 2013-14 UEFA Europa League competition? |
|---|---|
| Explanation | We first find the column of "home" and "away" and compare the outcome of "home" and "away" games. Then we find there is no correlation between "home" and "away" games. As a result, the answer is No. |
| Table | Read the table below regarding "yugoslavia national football team results" |

|   | date | city | opponent | results | type of game |
|--:|:---|:---|:---|:--:|:---|
| 0 | april 18 | belgrade | france | 1:0 | 1966 wcq |
| 1 | may 9 | belgrade | england | 1:1 | friendly |
| 2 | june 16 | oslo , norway | norway | 0:3 | 1966 wcq |
| 3 | september 4 | moscow , russia | ussr | 0:0 | friendly |
| 4 | september 19 | luxembourg | luxembourg | 5:2 | 1966 wcq |
| 5 | october 9 | paris , france | france | 0:1 | 1966 wcq |
| 6 | november 7 | belgrade | norway | 1:1 | 1966 wcq |

| Question | Did the Yugoslavia national football team play any games against teams outside of Europe in the table? Answer with only 'Yes' or 'No' that is most accurate and nothing else. |
|---|---|
| Explanation | |

---

**PAL**

| Instruction | Let's use Python to solve Table-based question answering. Here is an example how to do it, |
|---|---|
| Table | Tittle: "1982 all - ireland senior hurling championship" |

|   | rank | player | county | tally | total | matches | average |
|--:|:--:|:---|:---|:---|:--:|:--:|:--:|
| 0 | 1 | pádraig horan | offaly | 5 - 17 | 32 | 4 | 8 |
| 1 | 2 | billy fitzpatrick | kilkenny | 2 - 24 | 30 | 4 | 7.5 |
| 2 | 3 | tony o 'sullivan | cork | 0 - 28 | 28 | 4 | 7 |
| 3 | 4 | p j molloy | galway | 3 - 11 | 20 | 2 | 10 |
| 4 | 5 | christy heffernan | kilkenny | 3 - 9 | 18 | 4 | 4.5 |
| 5 | 5 | pat horgan | cork | 0 - 18 | 18 | 4 | 4.5 |

| Question | How many players in the 1982 all-Ireland senior hurling championship had a higher average score per game than the overall average score per game of the competition? |
|---|---|
| Code | |

```
import pandas as pd
# get the overall average score per game of the competition
overall_avg = df['average'].mean()
# filter the dataframe to only include players with a higher
average score per game than the overall average
higher_avg_df = df[df['average'] > overall_avg]
# count the number of players with a higher average score per game than the overall average
num_higher_avg_players = len(higher_avg_df)
# print the result
print("Number of players with a higher average score per game than the overall average: ", num_higher_avg_players)
```

| Instruction | "How about this question? " |
|---|---|
| Table | Tittle: "yugoslavia national football team results" |

|   | date | city | opponent | results | type of game |
|--:|:---|:---|:---|:--:|:---|
| 0 | april 18 | belgrade | france | 1:0 | 1966 wcq |
| 1 | may 9 | belgrade | england | 1:1 | friendly |
| 2 | june 16 | oslo , norway | norway | 0:3 | 1966 wcq |

| | 3 | september 4  | moscow , russia | ussr       | 0:0 | friendly  |
| | 4 | september 19 | luxembourg      | luxembourg | 5:2 | 1966 wcq  |
| | 5 | october 9    | paris , france  | france     | 0:1 | 1966 wcq  |
| | 6 | november 7   | belgrade        | norway     | 1:1 | 1966 wcq  |

**Question**    Did the Yugoslavia national football team play any games against teams outside of Europe in the table? Answer with only 'Yes' or 'No' that is most accurate and nothing else.

**Code**

---

**ReAct**

**Table**    Tittle: "1982 all - ireland senior hurling championship"

| |rank| player          | county   | tally  | total | matches | average |
|---:|---:|:----------------|:---------|:-------|:------|:--------|:--------|
| 0 | 1 | pádraig horan    | offaly   | 5 - 17 | 32    | 4       | 8       |
| 1 | 2 | billy fitzpatrick| kilkenny | 2 - 24 | 30    | 4       | 7.5     |
| 2 | 3 | tony o 'sullivan | cork     | 0 - 28 | 28    | 4       | 7       |
| 3 | 4 | p j molloy       | galway   | 3 - 11 | 20    | 2       | 10      |
| 4 | 5 | christy heffernan| kilkenny | 3 - 9  | 18    | 4       | 4.5     |
| 5 | 5 | pat horgan       | cork     | 0 - 18 | 18    | 4       | 4.5     |

**Question**    How many players in the 1982 all-Ireland senior hurling championship had a higher average score per game than the overall average score per game of the competition?

**Thought 1**    I need to first get the overall average score per game of the competition and then count the number of players with a higher average score per game than the overall average.

**Action 1**   
```
overall_avg = df['average'].mean()
print("The overall average score per game of the competition is " + overall_avg)
```

**Observation 1**    The overall average score per game of the competition is 6.916666666666667

**Thought 2**    Then I should filter the dataframe to only include players with a higher average score per game than the overall average which is 6.916666666666667 and count the length of the filtered dataframe.

**Action 2**   
```
higher_avg_df = df[df['average'] > overall_avg]
num_higher_avg_players = len(higher_avg_df)
print("Number of players with a higher average score per game than the overall average: ", num_higher_avg_players)
```

**Observation 2**    Number of players with a higher average score per game than the overall average: 4

**Thought 3**    Number of players with a higher average score per game than the overall average is 4. So the answer is 4.

**Action 3**    Finish. The answer is 4.

**Table**    Tittle: "yugoslavia national football team results"

| | date         | city            | opponent   | results | type of game |
|---:|:-------------|:----------------|:-----------|:--------|:-------------|
| 0 | april 18      | belgrade        | france     | 1:0     | 1966 wcq     |
| 1 | may 9         | belgrade        | england    | 1:1     | friendly     |
| 2 | june 16       | oslo , norway   | norway     | 0:3     | 1966 wcq     |
| 3 | september 4   | moscow , russia | ussr       | 0:0     | friendly     |
| 4 | september 19  | luxembourg      | luxembourg | 5:2     | 1966 wcq     |
| 5 | october 9     | paris , france  | france     | 0:1     | 1966 wcq     |
| 6 | november 7    | belgrade        | norway     | 1:1     | 1966 wcq     |

**Question**    Did the Yugoslavia national football team play any games against teams outside of Europe in the table? Answer with only 'Yes' or 'No' that is most accurate and nothing else.

---

**ARC**

**Instruction**    Based on the table, answer the following question ONLY using Python code. (use comments as the reasoning process)

**Table**    Tittle: "1982 all - ireland senior hurling championship"

| |rank| player          | county   | tally  | total | matches | average |
|---:|---:|:----------------|:---------|:-------|:------|:--------|:--------|
| 0 | 1 | pádraig horan    | offaly   | 5 - 17 | 32    | 4       | 8       |
| 1 | 2 | billy fitzpatrick| kilkenny | 2 - 24 | 30    | 4       | 7.5     |
| 2 | 3 | tony o 'sullivan | cork     | 0 - 28 | 28    | 4       | 7       |
| 3 | 4 | p j molloy       | galway   | 3 - 11 | 20    | 2       | 10      |
| 4 | 5 | christy heffernan| kilkenny | 3 - 9  | 18    | 4       | 4.5     |
| 5 | 5 | pat horgan       | cork     | 0 - 18 | 18    | 4       | 4.5     |

**Question**    How many players in the 1982 all-Ireland senior hurling championship had a higher average score per game than the overall average score per game of the competition?

**Code**    Zero-shot Generate through LLMs

**Table**    Tittle: "yugoslavia national football team results"

| | date         | city            | opponent   | results | type of game |
|---:|:-------------|:----------------|:-----------|:--------|:-------------|
| 0 | april 18      | belgrade        | france     | 1:0     | 1966 wcq     |
| 1 | may 9         | belgrade        | england    | 1:1     | friendly     |
| 2 | june 16       | oslo , norway   | norway     | 0:3     | 1966 wcq     |
| 3 | september 4   | moscow , russia | ussr       | 0:0     | friendly     |
| 4 | september 19  | luxembourg      | luxembourg | 5:2     | 1966 wcq     |
| 5 | october 9     | paris , france  | france     | 0:1     | 1966 wcq     |
| 6 | november 7    | belgrade        | norway     | 1:1     | 1966 wcq     |

**Question**    Did the Yugoslavia national football team play any games against teams outside of Europe in the table? Answer with only 'Yes' or 'No' that is most accurate and nothing else.

**Code**    In-context Generate through LLMs

**Code Output**    Code Output executed by external Python interpreter

**Answer**

---

Table 8: Prompt design for all baseline methods in the main experiment

| | |
|---|---|
| Task description | In table-based QA tasks, not all questions need to be answered. You should determine whether to answer according to table and commonsense knowledge, that is to judge if the following question is answerable or unanswerable. The definitions of unanswerable questions are as follows. |
| Definition of unanswerable questions | \| | Type of unanswerable \| Definition \|
\|—-:\|:————————————————\|:——————————————————————————————\|
\| 0 \| Out of scope \| Lacking essential information based on the given table. \|
\| 1 \| Hallucination \| The assumption in the question is invalid based on the table. \|
\| 2 \| Problematic \| The question itself contains logical error. \|
\| 3 \| Subjective \| The answer varies due to different metrics, algorithms, and criteria.\|
\| 4 \| Others \| Other types of questions that can not be labeled. \| |
| 4-shot | Read the table below regarding "1982 all - ireland senior hurling championship" to judge if the following question is answerable or unanswerable.
\| \|rank\| player \| county \| tally \| total \| matches \| average \|
\|—-:\|—-:\|:——————————————\|:—————\|:————\|:————\|—-:\|————-:\|
\| 0 \| 1 \| pádraig horan \| offaly \| 5 - 17 \| 32 \| 4 \| 8 \|
\| 1 \| 2 \| billy fitzpatrick \| kilkenny \| 2 - 24 \| 30 \| 4 \| 7.5 \|
\| 2 \| 3 \| tony o 'sullivan \| cork \| 0 - 28 \| 28 \| 4 \| 7 \|
\| 3 \| 4 \| p j molloy \| galway \| 3 - 11 \| 20 \| 2 \| 10 \|
\| 4 \| 5 \| christy heffernan \| kilkenny \| 3 - 9 \| 18 \| 4 \| 4.5 \|
\| 5 \| 5 \| pat horgan \| cork \| 0 - 18 \| 18 \| 4 \| 4.5 \| |
| Question | How many players in the 1982 all-Ireland senior hurling championship had a higher average score per game than the overall average score per game of the competition? |
| Answer: | answerable. (4.)
Read the table below regarding "g.d. estoril praia" to judge if the following question is answerable or unanswerable.
\| \| season \| competition \| round \| opponent \| home \| away \|
\|—-:\|:————-\|:——————————————-\|:—————-\|:————————\|:—————\|:—————\|
\| 0 \| 2013 - 14 \| uefa europa league \| 3q \| hapoel ramat gan \| 0 - 0 \| 1 - 0 \|
\| 1 \| 2013 - 14 \| uefa europa league \| play - off \| pasching \| 2 - 0 \| 2 - 1 \|
\| 2 \| 2013 - 14 \| uefa europa league \| group h \| sevilla \| 1 - 2 \| - \|
\| 3 \| 2013 - 14 \| uefa europa league \| group h \| slovan liberec \| - \| 1 - 2 \|
\| 4 \| 2013 - 14 \| uefa europa league \| group h \| freiburg \| - \| 1 - 1 \| |
| Question | Was there a correlation between GD Estoril Praia's performance in home games and away games during the 2013-14 UEFA Europa League competition? |
| Answer | answerable. (No.)
Read the table below regarding "1941 in brazilian football" to judge if the following question is answerable or unanswerable.
\| \| position \| team \| points \| played \| drawn \| lost \| against \| difference \|
\|-:\|————-:\|:——————————\|—-:\|—-:\|—-:\|—-:\|:————\|
\| 0 \| 1 \| corinthians \| 35 \| 20 \| 3 \| 1 \| 17 \| 44 \|
\| 1 \| 2 \| são paulo \| 31 \| 20 \| 5 \| 2 \| 32 \| 23 \|
\| 2 \| 3 \| palestra itália - sp \| 30 \| 20 \| 6 \| 2 \| 19 \| 25 \|
\| 3 \| 4 \| portuguesa \| 20 \| 20 \| 6 \| 7 \| 46 \| - 3 \|
\| 4 \| 5 \| santos \| 20 \| 20 \| 4 \| 8 \| 60 \| - 1 \|
\| 5 \| 6 \| são paulo railway \| 18 \| 20 \| 4 \| 9 \| 53 \| - 5 \|
\| 6 \| 7 \| hespanha \| 18 \| 20 \| 2 \| 10 \| 57 \| - 9 \|
\| 7 \| 8 \| portuguesa santista \| 15 \| 20 \| 7 \| 9 \| 43 \| - 2 \|
\| 8 \| 9 \| ypiranga - sp \| 14 \| 20 \| 4 \| 11 \| 52 \| - 3 \|
\| 9 \| 10 \| juventus \| 14 \| 20 \| 4 \| 11 \| 49 \| - 17 \|
\| 10 \| 11 \| comercial - sp \| 5 \| 20 \| 3 \| 16 \| 76 \| - 52 \| |
| Question | Are teams with higher points more likely to win the teams with lower points? |
| Answer | unanswerable.
Read the table below regarding "neo geo online collection" to judge if the following question is answerable or unanswerable.
(Due to its excessive size, the table is disregarded as it cannot be displayed) |
| Question | Does the length of the Japanese titles differ significantly between pre-2005 and post-2005 releases? |
| Answer | unanswerable. |
| Interface | Read the table below regarding "{title}" to judge if the following question is answerable or unanswerable.
{table}
Question: {question}
Answer with only "answerable" of "unanswerable" and no other outputs. (If the question is answerable, give your answers, otherwise answer with only "unanswerable".)
Answer: |

Table 9: Prompt design for binary classification and question-answering tasks. The prompt for the question-answering task is enclosed in parentheses and highlighted in green.

| Sub type | Definition and Template |
|---|---|
| Indexing | D: Mapping between the values in a specific column and the corresponding rows in a table
T: Find the row/columns of [ENTITY] |
| Filter | D: Retrieve data from a table based on specific conditions.
T: Filter the rows/columns based on the [ENTITY] |
| Grouping | D: Group data based on one or more columns/rows.
T: Filter the rows/columns based on the [ENTITY] |
| Sorting | D: Order data in a specific way.
T: Sort the rows/columns based on [ENTITY] |
| Grounding | D: Determining whether a given statement logically follows from a set of premises or background knowledge.
T: Group the rows/columns based on [ENTITY] |
| Auto-categorization | D: Categorizing or classifying information into predefined categories or groups based on its content.
T: The term of [ENTITY1] in the question is mapped to [ENTITY2] |
| Temporal Reasoning | D: Make presumptions about humans' knowledge of times, durations, and time intervals.
T: Based on the [ENTITY1] time, the temporal indicator is [ENTITY2] |
| Geographical/ Spatial Reasoning | D: Reasoning about Geographical/Spatial knowledge.
T: Conduct geographical/spatial reasoning on [ENTITY] |
| Aggregating | D: Combining multiple values into a single value to summarize data and make it easier to understand.
T: Conduct the aggregating operation of [ENTITY1] on the value of [ENTITY2]. |
| Arithmetic | D: Basic mathematical operations.
T: Conduct the arithmetic operation of [ENTITY1] on the value of [ENTITY2]. |
| Reasoning with Quantifiers | D: The process of making logical and mathematical inferences from statements that contain quantifiers.
T: Conduct the reasoning with quantifiers of [ENTITY1] on the domain of discourse of [ENTITY2] |

Table 10: The definitions (D) and templates (T) of reasoning and operations types in our proposed taxonomy.

**Question:** Are there any anomalous observations in terms of popular vote percentage for candidates in the grassroots party?

**Answer:** [Type the answer]

Bridging: requires finding the first-hop evidence to find the second-hop one.

Intersection: requires finding an entity that satisfies two independent conditions.

Comparison: requires comparing the property of two different entities.

**How do sub-questions compose?**
- ● Bridging
- ○ Intersection
- ○ Comparison
- ○ Other

Explicit: The question decomposition can be written with a vocabulary limited to words from the questions, their inflections, and function words.

Implicit: otherwise.

**Is the question implicit or explicit?**
- ● Implicit
- ○ Explicit

**Operation Types**

a. Indexing:

    Definition: mapping between the values in a specific column and the corresponding rows in a table.

    Examples:  For the data in the second row and third column, …

b. Filter:

    Definition: retrieve data from a table based on specific conditions.

    Examples: find the events happened in 2013 , …

c. Grouping:

    Definition: group data based on one or more columns/rows.

    Examples: For male athletes , …

d. Sorting:

    Definition: order data in a specific way.

    Examples: select the player who have the third highest scores,  …

**Reasoning Types**

a. Aggregating :

    Definition: combining multiple values into a single value to summarize data and make it easier to understand.

    Examples: e.g., sum(), average(), min(), max(), …

b. Arithmetic :

    Definition: basic mathematical operations

    Examples: e.g., + - * / , <, >

c. Grounding:

    Definition: Determining whether a given statement logically follows from a set of premises or background knowledge.

    Examples: Turnover exceeds costs -> make a profit

d. Auto-categorization:

    Definition: Categorizing or classifying information into predefined categories or groups based on its content and context.

    Examples: yellow, blue, green -> colors

e. Temporal Reasoning:

    Definition: Make presumptions about humans' knowledge of times, durations and time intervals.

    Examples: In the next two years, …

f. Geographical/ Spatial reasoning:

    Examples: e.g., For palyers from Asia, …

g. Reasoning with Quantifiers"

    Definition: The process of making logical and mathematical inferences from statements that contain quantifiers.

    Examples:  e.g., Are all dogs mammals?  (universal quantifier)  There exists a dog that is friendly. (existential quantifier)

f. Other commonsense reasoning: e.g., social reasoning,

**Step 1**
- ● Indexing
- ○ Filter
- ○ Grouping
- ○ Sorting
- ○ Grounding
- ○ Auto-categorization
- ○ Temporal Reasoning
- ○ Geographical/Spatial Reasoning
- ○ Other Commonsense Reasoning
- ○ Aggregating
- ○ Arithmetic
- ○ Reasoning with Quantifiers

Submit

**For the operation of Indexing, fill the template: find the row/columns of [ENTITY]** [ENTITY]

**Step 2**
- ○ Indexing
- ○ Filter
- ● Grouping
- ○ Sorting
- ○ Grounding
- ○ Auto-categorization
- ○ Temporal Reasoning
- ○ Geographical/Spatial Reasoning
- ○ Other Commonsense Reasoning
- ○ Aggregating
- ○ Arithmetic
- ○ Reasoning with Quantifiers

Submit

**For the operation of Grouping, fill the template: Group the rows/columns based on [ENTITY]** [ENTITY]

**Step 3**
- ○ Indexing
- ○ Filter
- ○ Grouping
- ○ Sorting
- ○ Grounding
- ○ Auto-categorization
- ○ Temporal Reasoning
- ○ Geographical/Spatial Reasoning
- ○ Other Commonsense Reasoning
- ● Aggregating
- ○ Arithmetic
- ○ Reasoning with Quantifiers

Submit

**For the reasoning of Aggregating, fill the template: Conduct the aggregating**

**operation of [ENTITY1] on the value of [ENTITY2].** SUM ∨ [ENTITY]

**Step 4**
- ○ Indexing
- ○ Filter
- ○ Grouping
- ○ Sorting
- ○ Grounding
- ○ Auto-categorization
- ○ Temporal Reasoning
- ○ Geographical/Spatial Reasoning
- ○ Other Commonsense Reasoning
- ○ Aggregating
- ○ Arithmetic
- ● Reasoning with Quantifiers

Submit

**For the reasoning of Reasoning with Quantifiers, fill the template: Conduct the reasoning**

**with quantifiers of [ENTITY1] on the domain of discourse of [ENTITY2].** universal ∨ [ENTITY]

Submit

Figure 8: Our detailed data annotation interface.