# OpenReview forum: "CRT-QA: A Dataset of Complex Reasoning Question Answering over Tabular Data"
_EMNLP/2023/Conference — EMNLP 2023 Main_

### Official Review · Reviewer_dBCv · 2023-08-04

**Soundness:** 4

**Excitement:**

4: Strong: This paper deepens the understanding of some phenomenon or lowers the barriers to an existing research direction.

**Paper Topic And Main Contributions:**

This paper introduces a new Question Answering dataset over Tabular data which requires complex reasoning, CRT-QA. This dataset includes fine-grained information such as type of operation and type of reasoning, which can be useful for diagnosis models. They show that CRT-QA is a difficult task for the large language model.
They propose a method that utilizes a language model to translate a question into code, which can then use an external python library tool to answer the question.


**Reasons To Accept:**

1. The author introduces a more complex reasoning QA dataset over tabular data, which is created by human-machine-in-the-loop method. Additional meta-data is given in the dataset which can be beneficial for the model diagnosis.
2. They proposed a method that can achieve better performance than zero-shot models or few-shot learning models. Their methods have advantage in terms of interpretability and also get rid of human crafted few-shot examples.


**Reasons To Reject:**

Overall, this is a good resource for the community and the proposed method is intuitive and achieves good performance. But I think there are few things can be done to make the paper better.

1. The dataset is a test-only dataset, since most of the models are not trained on table corpus, it might be reasonable that such language models can not do well on table dataset. The author did not test a table-oriented model such as tableBERT on the proposed dataset, which can be interesting to see.
2. To show that CRT-QA is more complex than the previous benchmark, the author might need to show that the same language model can easily achieve better results on previous dataset.
3. It is unclear that Table 5 is the result from which language model. And also, is this result just evaluated based on the unanswerable/indeterminate subset? If so, why not do the evaluation on the mix of unanswerable/indeterminate and normal questions? Will the model say a normal question is unanswerable/indeterminate?


**Reproducibility:**

3: Could reproduce the results with some difficulty. The settings of parameters are underspecified or subjectively determined; the training/evaluation data are not widely available.

**Reviewer Confidence:**

4: Quite sure. I tried to check the important points carefully. It's unlikely, though conceivable, that I missed something that should affect my ratings.

**Typos Grammar Style And Presentation Improvements:**

Naming: “ARC” has been taken as the name for the dataset AI2 Reasoning Challenge[1], to avoid duplication, maybe the author can have a different name for the proposed model.

[1]Think you have Solved Question Answering? Try ARC, the AI2 Reasoning Challenge

---

> ### Author Rebuttal · Authors · 2023-08-28
>
> Thank you for your meaningful feedback and comments on our work. We're glad you recognized the **uniqueness of our complex reasoning** TQA dataset and the value of the additional meta-data we provided. Your observation about our method's advantages in terms of **performance, interpretability, and the reduction of human-crafted few-shot examples** is on point. We appreciate your thorough review and are open to any additional suggestions or insights you may have.
>
> 1. W1: Test on a table-oriented model
>
>     As our work focuses on investigating LLMs’ reasoning abilities over tabular data, we do not incorporate finetune-based models in our experiments. In response to your comments, we further conduct **additional experiments** with **two types** of fine-tuned models on CRT-QA. First, we test our dataset on **TaPex** [1], one of the **SOAT tabular pre-trained models** based on BART. Then, we test a **SOTA code-completion model, Starcoder** [2]. We conduct zero-shot prompting and PAL for TaPex and code-completion models respectively.
>
>
>     | Model (Method) | Index | Sort | Group | Filter | GRO | CAT | TEM | AGG | ARI | SPA | QUA | OTH | Overall |
>     | --- | --- | --- | --- | --- | --- | --- | --- | --- | --- | --- | --- | --- | --- |
>     | Tapex (zero-shot) | 35.74 | 22.5 | 45.26 | 32.38 | 27.66 | 0.00 | 25.93 | 28.68 | 27.31 | 26.32 | 54.44 | 37.70 | 35.05 |
>     | Starcoder (PAL) | 26.15 | 25.00 | 26.95 | 12.70 | 26.95 | 22.22 | 25.93 | 14.91 | 18.49 | 5.26 | 40.56 | 26.67 | 23.64 |
>
>     From the table, we find that both Tapex and Starcoder can not reach the performance of ChatGPT. Besides, Tapex performs much better than Starcoder. One of the reasons may be that Starcoder was not trained specifically on tabular data.
>
> 2. W2: To prove the challenge of CRT-QA, the author might need to show that the same LLM can easily achieve better results on the previous dataset.
>     - In response to your comments, we have conducted **additional experiments** with GPT-4 on **two widely used TQA datasets**, WikiTableQuestions (**WTQ**) [3] and **HiTab** [4]. WTQ is one of the earliest and one of the most widely-used TQA datasets which contains 4344 tabular questions. We use the same prompt design as Chen et al. [5] for zero-shot and CoT experiments. The more recent HiTab contains 1584 tabular QA in the test set and all of their tables are complex hierarchical tables with multi-level headers. The experiment results and previous SOTA performance are listed as follows:
>
>         |  | GPT-4 Zero-shot | GPT-4 CoT | Reported SOTA |
>         | --- | --- | --- | --- |
>         | WikiTableQuestions | 71.39 | 73.76 | 52.7 [6] |
>         | HitTab  | 87.84 | 91.36 | 40.7 [4] |
>         | Averaged EM difference between CRT-QA | 33.41 | 26.24 | - |
>
>         From the table, we find that GPT-4 can achieve **much better results** on previous TQA datasets than CRT-QA **even without external tools**, indicating the challenge of our proposed dataset. Besides, GPT-4 can easily surpass previous SOTA model on these datasets.
>
> 3. W3: Details about Figure 5.
>
>     The experiment is tested on the whole dataset, including unanswerable/indeterminate and "normal" questions. We conducted tests on ChatGPT using a few-shot prompt and a temperature setting of 0 in order to investigate the language model's capabilities preliminarily to determine whether to choose to answer. Detailed prompt design can be found in Table 7 in the Appendix. The confusion matrix of binary classification is `TN:202 FP:48 FN:270 TP:474`, while `TN:206 FP:44 FN:176 TP:568` for question answering.
>
>     ChatGPT sometimes considers a “normal” question to be unanswerable/indeterminate both in binary classification and question answering. We will provide more details about this experiment in camera-ready version.
>
>
> [1] Liu, Qian, et al. "TAPEX: Table Pre-training via Learning a Neural SQL Executor." *International Conference on Learning Representations*. 2021.
>
> [2] Li, Raymond, et al. "StarCoder: may the source be with you!." arXiv e-prints (2023): arXiv-2305.
>
> [3] Panupong Pasupat and Percy Liang. 2015. Compositional Semantic Parsing on Semi-Structured Tables. In Proceedings of the 53rd Annual Meeting of the Association for Computational Linguistics and the 7th International Joint Conference on Natural Language Processing (Volume 1: Long Papers), pages 1470–1480, Beijing, China. Association for Computational Linguistics.
>
> [4] Zhoujun Cheng, Haoyu Dong, Zhiruo Wang, Ran Jia, Jiaqi Guo, Yan Gao, Shi Han, Jian-Guang Lou, and Dongmei Zhang. 2022. HiTab: A Hierarchical Table Dataset for Question Answering and Natural Language Generation. In Proceedings of the 60th Annual Meeting of the Association for Computational Linguistics (Volume 1: Long Papers), pages 1094–1110, Dublin, Ireland. Association for Computational Linguistics.
>
> [5] Wenhu Chen. 2023. Large Language Models are few(1)-shot Table Reasoners. In Findings of the Association for Computational Linguistics: EACL 2023, pages 1120–1130, Dubrovnik, Croatia. Association for Computational Linguistics.
>
> [6] Tao Yu, Chien-Sheng Wu, Xi Victoria Lin, Bailin Wang, Yi Chern Tan, Xinyi Yang, Dragomir R Radev, Richard Socher, and Caiming Xiong. 2021. Grappa: Grammar-augmented pre-training for table semantic parsing. In ICLR.

---

### Official Review · Reviewer_ejgp · 2023-08-05

**Soundness:** 4

**Excitement:**

4: Strong: This paper deepens the understanding of some phenomenon or lowers the barriers to an existing research direction.

**Missing References:**

In LLM with tools section of related work: Decomposed Prompting (https://arxiv.org/abs/2210.02406) which also uses tools (or modules) for complex tasks.

**Paper Topic And Main Contributions:**

This work proposes a new TableQA dataset called CRT-QA. Unlike past datasets,

1. that mainly have explicit reasoning questions (having operations like select, filter, etc are easily identifiable from question words), this work introduces questions that are more implicit in nature (requiring informal/commonsense reasoning).
2. this dataset has unanswerable or interminate questions to assess models' answerability prediction.
3. this dataset has a fine-grained annotation of reasoning paths, subquestions, and answers required to answer complex questions.

To build this dataset, they prompt LLMs to generate questions from tables, and then have humans give feedback to improve the question. They have humans annotate the reasoning paths using templates with slots.

This work also introduces a model, ARC, that essentially works by iterative execution of pandas code. They bootstrap demonstrations based on the dev set instead of hand-crafting them. Then they compare it with many other few-shot LLM strategies and show it works the best (~60%), which still doesn't solve the dataset.

**Questions For The Authors:**

Ignore if space constrained (won't influence rating), prioritize weaknesses:

- A. How are the reasoning path annotations used in ARC prompts, if at all? Although these reasoning paths are useful, I'm not sure how and where you are using them. Perhaps, only in building CoT prompts for non-ARC baselines?

**Reasons To Accept:**

- This new dataset has multiple novel and useful features over the existing TableQA datasets -- implicit questions, reasoning paths, and unanswerable questions. Each of these is an important contribution and can be beneficial to people working in this TableQA field.

- Although it's not a modeling paper, this paper also introduces an interesting approach that solving the task, which can be applied easily to other TableQA datasets as well. The authors also compare their modeling approach with several other few-shot approaches across different reasoning types, providing a good analysis. (Although not necessary, it would be useful to compare ARC's performance with other approaches on existing TableQA datasets, perhaps in the appendix)

- The paper is well written.

**Reasons To Reject:**

__It's unclear what's the (human) UB or solvability of this dataset.__

First of all, it's unclear how a ground-truth answer is generated. Perhaps, reasoning path annotation already involves writing the final answer, but I'm not sure. But importantly, the information about assessing the quality of the dataset solvability or (human) upper bound is missing. This is especially important as the questions are partly autogenerated. On a related note, it'd also be useful to discuss how the difficulty of this dataset compares with other datasets for the current best models.

__There is no fine-tuned model baseline for this new dataset.__

Although I understand people are mainly interested in zero/few-shot settings these days, assessing the solvability of tasks with finetuning can also be useful for people who are interested in it.

Please correct me if I'm missing something here. I can increase the score post-rebuttal.

**Reproducibility:**

5: Could easily reproduce the results.

**Reviewer Confidence:**

4: Quite sure. I tried to check the important points carefully. It's unlikely, though conceivable, that I missed something that should affect my ratings.

**Typos Grammar Style And Presentation Improvements:**

It was not clear to me what you meant by "(for past datasets) definition of reasoning does not align with the reasoning ability of LLMs" in related work and also in the intro. I only understood it in 183-185. This is a good point, consider bringing it introduction or related work to better distinguish your work early enough.

There a capitalization mistakes at 2-3 places (e.g., L564). Consider proofreading again.

L060: on -> of

---

> ### Author Rebuttal · Authors · 2023-08-28
>
> Thank you for your thorough review of our paper. The **novel and useful features** of **implicit questions, reasoning paths, and unanswerable questions** indeed set our dataset apart from the existing ones, and we are glad to see that you find these contributions **significant**.
>
> - W1: It's unclear what's the (human) UB or solvability of this dataset.
>     - The **inter-annotator agreement** reported in **Table 2** can reflect the human upper bound. The over **93% of the inter-annotator agreement** indicates our dataset’s considerable solvability for humans. During the data construction process, we have **separated the unanswerable and Indeterminate questions** which can alleviate the ambiguity to ensure the solvability for "normal" questions.
>
> - W2: no fine-tuned model baseline for this new dataset.
>     - As our work focuses on investigating LLMs’ reasoning abilities over tabular data, we do not incorporate finetune-based models in our experiments. In response to your comments, we further conduct **additional experiments** with **two types of fine-tuned models** on CRT-QA. First, we test our dataset on TaPex [1], one of the **SOAT tabular pre-trained models** based on BART. Then, we test a SOTA **code-completion model**, Starcoder [2]. We conduct zero-shot prompting and PAL for TaPex and code-completion models respectively.
>
>     | Model (Method) | Index | Sort | Group | Filter | GRO | CAT | TEM | AGG | ARI | SPA | QUA | OTH | Overall |
>     | --- | --- | --- | --- | --- | --- | --- | --- | --- | --- | --- | --- | --- | --- |
>     | Tapex (zero-shot) | 35.74 | 22.5 | 45.26 | 32.38 | 27.66 | 0.00 | 25.93 | 28.68 | 27.31 | 26.32 | 54.44 | 37.70 | **35.05** |
>     | Starcoder (PAL) | 26.15 | 25.00 | 26.95 | 12.70 | 26.95 | 22.22 | 25.93 | 14.91 | 18.49 | 5.26 | 40.56 | 26.67 | **23.64** |
>
>     From the table, we find that both Tapex and Starcoder can not reach the performance of ChatGPT. Besides, Tapex performs much better than Starcoder. One of the reasons may be that Starcoder was not trained specifically on tabular data.
>
> - Q1: How are the reasoning paths used?
>     - As Review r1r7 mentioned, "our TQA dataset’s uniqueness in multi-step reasoning and reasoning chain annotations". Evaluating LLMs’ reasoning path is an **important but challenging** task and the current approaches are **most goal-oriented**, which is **far from comprehensive**. Our annotation of fine-grained reasoning paths is geared towards providing a reference to evaluate LLM’s reasoning process (**process-oriented**). For example, we use human-annotated reasoning paths to evaluate LLM’s reasoning process in vanilla prompting baselines **(lines 343-348)** and find that LLMs can often generate correct reasoning plans but are **unable to correctly execute them**, inspiring us to **incorporate external tools** to augment LLMs.
> - Typos Grammar Style And Presentation Improvements:
>     - T1: What does "(for past datasets) definition of reasoning does not align with the reasoning ability of LLMs” mean?
>         - It has the exact same meaning as lines 183-185. As you suggested, we will bring it introduction or related work in the camera-ready version.
>
> [1] Liu, Qian, et al. "TAPEX: Table Pre-training via Learning a Neural SQL Executor." *International Conference on Learning Representations*. 2021.
>
> [2] Li, Raymond, et al. "StarCoder: may the source be with you!." arXiv e-prints (2023): arXiv-2305.

---

### Official Review · Reviewer_r1r7 · 2023-08-11

**Soundness:** 4

**Excitement:**

4: Strong: This paper deepens the understanding of some phenomenon or lowers the barriers to an existing research direction.

**Paper Topic And Main Contributions:**

The paper introduces the CRT-QA dataset, curated for facilitating complex reasoning over tabular data. This contribution is a commendable addition to the table question answering (TQA) research domain. Furthermore, the authors provide a rigorous baseline evaluation on this novel dataset, demonstrating the challenges posed by the dataset even to contemporary large language models.

**Questions For The Authors:**

- From Figure 3, we see that ARC is based on Python Pandas code generation. Will SQL be a better choice for this table question answering?
- Could you provide more baseline results and analysis on CRT-QA? e.g., open-source code generation models (starcoder, santacoder,…) or finetuned language models.

**Reasons To Accept:**

The paper's significance in the TQA domain is evident due to the following:

- The CRT-QA is a pioneering dataset tailored for intricate reasoning over tabular data, highlighting the existing challenges for state-of-the-art pretrained language models.
- CRT-QA can motivate future AI research in data science. It distinguishes itself from other datasets, as many lack multi-step reasoning annotations. CRT-QA incorporates a reasoning chain, delineating the process to derive answers from the table – a valuable feature in the current TQA landscape.
- The paper showcases clarity in its writing and is methodically structured, facilitating a coherent understanding.

**Reasons To Reject:**

While the paper is commendable, there are some aspects that require further consideration:

- A significant proportion of the baselines leverage the ChatGPT series models. To provide a holistic understanding, the inclusion of results from other architectures, such as finetuned language models and end-to-end models, would be beneficial.
- Exact Match (EM) might not provide a comprehensive evaluation of a model's performance on CRT-QA. The authors could enrich their analysis by incorporating additional evaluation metrics.

**Reproducibility:**

5: Could easily reproduce the results.

**Reviewer Confidence:**

4: Quite sure. I tried to check the important points carefully. It's unlikely, though conceivable, that I missed something that should affect my ratings.

---

> ### Author Rebuttal · Authors · 2023-08-28
>
> Thanks for your insightful and positive feedback! We are encouraged that you found our proposed CRT-QA **pioneering and potential impact** on **future AI research in data science**. We are also glad that you highlighted our TQA dataset’s **uniqueness in multi-step reasoning and reasoning chain annotations**, which differentiate ours from others in the TQA landscape.
>
> - W1: Lack of traditional fine-tuned models’ performance.
>     - As our work focuses on investigating LLMs’ reasoning abilities over tabular data, we do not incorporate finetune-based models in our experiments. In response to your comments, we further conduct **additional experiments** with **two types of fine-tuned models** on CRT-QA. First, we test our dataset on TaPex [1], one of the **SOAT tabular pre-trained models** based on BART. Then, we test a SOTA **code-completion model**, Starcoder [2]. We conduct zero-shot prompting and PAL for TaPex and code-completion models respectively.
>
>     | Model (Method) | Index | Sort | Group | Filter | GRO | CAT | TEM | AGG | ARI | SPA | QUA | OTH | Overall |
>     | --- | --- | --- | --- | --- | --- | --- | --- | --- | --- | --- | --- | --- | --- |
>     | Tapex (zero-shot) | 35.74 | 22.5 | 45.26 | 32.38 | 27.66 | 0.00 | 25.93 | 28.68 | 27.31 | 26.32 | 54.44 | 37.70 | **35.05** |
>     | Starcoder (PAL) | 26.15 | 25.00 | 26.95 | 12.70 | 26.95 | 22.22 | 25.93 | 14.91 | 18.49 | 5.26 | 40.56 | 26.67 | **23.64** |
>
>     From the table, we find that both Tapex and Starcoder can not reach the performance of ChatGPT. Besides, Tapex performs much better than Starcoder. One of the reasons may be that Starcoder was not trained specifically on tabular data.
>
> - W2: Only the EM score for evaluation may not be comprehensive enough.
>     - As described in the section of **Limitation (lines 552-567)**, we acknowledge that the EM score has its own limitations. For example, EM fails in many cases like “15” and “fifteen”. As a result, we **manually calibrate (see lines 553-554)** such cases. In the section of Limitation, we also further discuss the limitations and challenges of current metrics on free-form answer-generation tasks. Generally, the EM score is suitable for our task because, during the data construction process, we **only** select the questions that can be answered **within several words without ambiguity**, which can be easily evaluated using EM for most cases.
> - Q1: Why use Python Pandas instead of SQL  for code generation? Will SQL be a better choice?
>     - Our ARC framework can **easily incorporate** **any other programming language** such as SQL. In our experiments, we opted for Python Pandas because it is much more **flexible and versatile** than SQL. For example, when processing comparison tasks, SQL typically outputs a concrete query result or numerical value instead of a boolean value (a judgment of true or false).
>
>         | Name | Age | Gender |
>         | --- | --- | --- |
>         | John | 25 | Male |
>         | Sarah | 30 | Female |
>         | Alex | 25 | Male |
>
>         For example, if we have a table with columns Name, Age, and Gender, and we want to know which gender has a higher average age, we can use the following SQL query: `SELECT Gender, AVG(Age) as AvgAge FROM table_name GROUP BY Gender`. This will return a result set with two rows of data: one row displaying the average age of males and another row displaying the average age of females. However, further logical processing is needed to determine the final result. Although our proposed ARC can handle it in the last step of Iterative LLM calling with code output, **other code-based baselines** such as PAL **can not deal with such cases**.
>
> - Q2: Can you provide more baseline results and analysis on CRT-QA?
>     - Please refer to the response to W1.
>
> [1] Liu, Qian, et al. "TAPEX: Table Pre-training via Learning a Neural SQL Executor." *International Conference on Learning Representations*. 2021.
>
> [2] Li, Raymond, et al. "StarCoder: may the source be with you!." arXiv e-prints (2023): arXiv-2305.

---

### Meta-Review · Area_Chair_fnLz · 2023-09-12

**Recommendation:** 5

**Metareview:**

This paper presents a noteworthy addition to the field of table question answering (TQA) research. The authors present a new dataset and offer a comprehensive baseline evaluation of this innovative dataset, illustrating the substantial challenges it presents even to modern large language models. It stands out from other datasets, as many lack annotations for multi-step reasoning. CRT-QA incorporates a reasoning chain, delineating the step-by-step process required to derive answers from the table, which is a valuable feature in the current TQA landscape.

While not primarily a modeling paper, this work also introduces an intriguing approach to solving the task, which can be readily applied to other TableQA datasets. Additionally, the authors compare their modeling approach with several other few-shot approaches, encompassing various reasoning types, and providing a thorough analysis.

The authors have effectively addressed the reviewers' concerns by promptly conducting additional experiments to furnish more evidence, thereby offering a more holistic understanding of their proposed model.

---

### Decision · Program_Chairs · 2023-10-07

**Decision:**

Accept-Main

**Comment:**

This paper presents a noteworthy addition to the field of table question answering (TQA) research. The authors present a new dataset and offer a comprehensive baseline evaluation of this innovative dataset, illustrating the substantial challenges it presents even to modern large language models. It stands out from other datasets, as many lack annotations for multi-step reasoning. CRT-QA incorporates a reasoning chain, delineating the step-by-step process required to derive answers from the table, which is a valuable feature in the current TQA landscape.

While not primarily a modeling paper, this work also introduces an intriguing approach to solving the task, which can be readily applied to other TableQA datasets. Additionally, the authors compare their modeling approach with several other few-shot approaches, encompassing various reasoning types, and providing a thorough analysis.

The authors have effectively addressed the reviewers' concerns by promptly conducting additional experiments to furnish more evidence, thereby offering a more holistic understanding of their proposed model.